# Comparison of Whole Blood Cryopreservation Methods for Extensive Flow Cytometry Immunophenotyping

**DOI:** 10.3390/cells11091527

**Published:** 2022-05-02

**Authors:** Valentina Serra, Valeria Orrù, Sandra Lai, Monia Lobina, Maristella Steri, Francesco Cucca, Edoardo Fiorillo

**Affiliations:** 1Institute for Genetic and Biomedical Research, National Research Council (CNR), Cittadella Universitaria di Monserrato, 09042 Cagliari, Italy; valeria.orru@irgb.cnr.it (V.O.); sandra.lai@irgb.cnr.it (S.L.); monia.lobina@irgb.cnr.it (M.L.); maristella.steri@irgb.cnr.it (M.S.); fcucca@uniss.it (F.C.); edoardo.fiorillo@irgb.cnr.it (E.F.); 2Department of Biomedical Sciences, University of Sassari, 07100 Sassari, Italy

**Keywords:** whole blood, cryopreservation, immunophenotyping, flow cytometry, method comparison

## Abstract

Fresh blood immunophenotyping by flow cytometry, based on the reliable simultaneous detection of several markers in a cell, is the method of choice to study the circulating human immune system. Especially in large and multicenter studies, high sample quality is difficult to achieve, and adequate collection and storage of samples with fine-tuned whole blood cryopreservation is mandatory. Here, we compared the quality of immunophenotypic data obtained from fresh blood with those obtained after five cryopreservation methods by quantifying the levels of 41 immune cell populations. They comprised B and T lymphocyte subsets and their maturation stages, as well as monocytes and granulocytes. Three methods used fixative solutions and two other methods used dimethyl sulfoxide solutions to preserve cell viability. The fixative methods prevented detection of markers critical for identification of B and T cell subsets, including CD27, CXCR3, and CCR6. The other two methods permitted reliable discrimination of most immune-cell populations in thawed samples, though some cell frequencies varied compared to the corresponding fresh sample. Of those two methods, the one preserving blood in media containing dimethyl sulfoxide produced results that were most similar to those with fresh samples.

## 1. Introduction

Flow cytometry, technology that allows multiparametric identification and quantification of cells in fluid suspension, has been successfully used in recent decades to dissect the variability of the human immune system, starting from only a few drops of blood [1,2,3]. Fresh whole blood (WB) or peripheral blood mononuclear cells (PBMCs) are the biological materials of choice for immunophenotyping, providing a “snapshot” of leukocyte populations in the bloodstream. However, when large numbers of samples are obtained at the same time, and especially at sites distant from the flow cytometry facility, PBMC cryopreservation has been used as a reliable alternative source of material. Indeed, thawed PBMCs have already been used for massive immunophenotyping [4], but the time needed for their isolation and storage, and the accompanying complete loss of neutrophils, are important disadvantages that limit their utility.

Such problems have been compounded by the need to assess the influence of genetic variants on immunophenotypic variation, or the role of environmental factors on the human immune response to pathogens in health and disease. Such challenging studies, which require the collection and analysis of many thousands of individuals, are further complicated by the need to ensure high-quality data acquisition. Whole blood sample cryopreservation that allows rapid leukocyte storage would be the optimal way to resolve these issues.

Previous studies have tested commercially available kits to cryopreserve WB for immunophenotyping. In 2018, Olin and colleagues designed a mass cytometry panel comprising 38 antibodies to study immune system variations in cryopreserved WB from 100 newborns. They used two of the three fixative methods (CytoDelics and Smart tube) we are comparing in this manuscript; however, they did not show specific data regarding eventual differences in blood cryopreserved with the two methods they applied [5]. In 2019, Lee and collaborators used frozen blood (preserved with Smart tubes) for immune characterization by flow cytometry [6], but like Olin et al., they provide no data comparing fresh and frozen samples. More recently, Braudeau and colleagues compared immunophenotypes obtained from fresh blood with matched cryopreserved samples, using a commercially available solution, Cryostor [7]. They comprehensively dissected intracellular and surface markers covering both innate and adaptive immunity but focusing on only one method of cryopreservation. As an alternative to commercial blood cryopreservation kits, medium containing dimethyl sulfoxide (DMSO) has been used to cryopreserve WB for subsequent immunophenotyping by flow cytometry [8,9,10,11]. However, despite the use of various methods in different projects, systematic studies that compare the efficacy of the currently available blood cryopreservation methods are still missing.

Here, we compared the quality of immunophenotypic data obtained from fresh blood with that obtained by five cryopreservation methods by dissecting 41 leukocyte subsets, including B and CD4 T cell subtypes. Once we identified the most reliable freezing method, we tested an additional 11 cell types, including monocyte, Natural Killer (NK), and cytotoxic T subsets (Figure 1).

We found that cryopreserved whole blood can indeed sustain reliable large-scale flow cytometry immunophenotyping, but the proper selection of cryopreservation method is critical to obtain data comparable with those obtained from fresh blood.

## 2. Materials and Methods

### 2.1. The SardiNIA Dataset

Whole blood was collected from the SardiNIA project cohort, a longitudinal study begun in 2001, comprising about 8000 general population volunteers (57% females, 43% males), ranging from 18 to 102 years, followed-up at 5-year intervals since then. All volunteers live on the central east coast of Sardinia, Italy. Volunteers were extensively characterized for clinical and biochemical traits and genetically characterized for about 26 million variants. Five freezing methods were applied to a subset of five individuals who had no overt or reported pathological condition (Table 1). Furthermore, 3757 cohort individuals were profiled for more than 700 immune traits [1,3,12], providing reference data for the present study.

Blood collection was performed between 8:00 a.m. and 9:00 a.m. in BD Vacutainer heparinized blood collection tubes (BD Life Sciences, Franklin Lakes, NJ, USA, #367878), and was processed within 2 h for fresh blood staining and cryopreserved using the methods described in Section 2.2.

### 2.2. WB Cryopreservation and Thawing

WB from SardiNIA project volunteers was frozen, then thawed and stained as follows. Importantly, for commercial cryopreservation methods, freezing and thawing procedures were performed following manufacturer instructions.

#### 2.2.1. Method A: Proteomic Stabilizer from Smart Tube (SmT) (# MTS1P 100/CS, San Carlos, CA, USA)

One volume of WB was added to a SmT, capped, and inverted. Blood was then activated by breaking the ampoule inside the tube containing the proteomic stabilizer, which is sealed under an inert atmosphere. SmTs were then incubated for 10 min at room temperature (RT). SmTs were placed in a specific rack and transferred to −80 °C.

Samples were thawed at +4 °C for 30 min. Blood was then filtered with a 70 µm filter and the cells were lysed in two 10 min treatments at RT with Thaw Lyse Buffer. Samples were then washed once with BD Stain Buffer (BD Biosciences, #554657) for subsequent staining.

#### 2.2.2. Method B: CytoDelics Whole Blood Cell Stabilizer from CytoDelics (CyD, #hC001-1000)

One volume of WB was stabilized with an equal volume of CytoDelic Stabilizer (CytoDelics, #hC001-1000-S01), mixed and incubated for 10 min at RT. Stabilized blood was then aliquoted into cryovials and stored at −80 °C.

After thawing at +37 °C, one volume of blood was fixed with 10 volumes of Fixation Buffer (CytoDelics, #hC001-1000-F01) for 15 min at RT. Samples were then lysed with 40 volumes of Lysis Buffer (CytoDelics, #hC001-1000-L01) for 20 min at RT and washed with an equal volume of Wash Buffer (CytoDelics, #hC001-1000-W01) for subsequent staining.

#### 2.2.3. Method C: Freezing Mix (FM) Solution of 10% DMSO (Sigma-Aldrich, St. Luis, MO, USA, #D2650) Diluted in RPMI 1640 Medium (Lonza,#BE12-167F)

One volume of WB was mixed with one volume of a freezing mix consisting of 20% of DMSO diluted into RPMI 1640 medium, resulting in a final DMSO concentration of 10%. The diluted sample was then stored at −80 °C.

The FM sample was thawed for 15 min on ice and washed once with Phosphate Buffered Saline 1X (PBS, Sigma-Aldrich, #D8537), then stained.

#### 2.2.4. Method D: Red Blood Cell Depletion Kit, HetaSep (HeS) from Stemcells (#07906)

Five volumes of WB were mixed with 1 volume of HetaSep solution and mixed by pipetting. Each sample was then incubated at +37 °C for a time adequate for erythrocyte sedimentation. The leukocyte-enriched supernatant was then harvested and washed with PBS. Cold Cryostor CS10 (Sigma-Aldrich, #C2874) containing DMSO was added to leukocytes and the mixture incubated at +4 °C for 10 min. Samples were then aliquoted into cryovials and stored at −80 °C.

HeS samples were thawed at +37 °C, washed with PBS, and stained.

#### 2.2.5. Method E: TransFix/EDTA Vacuum Blood Collection Tubes from Cytomark (TVTs, #TVT-03-50)

WB collected in TVTs was immediately stored at −80 °C and tubes placed in stand inclined at a 45 degree angle [13]. Each TVT contains a solution of TransFix (cell preservative) and K3EDTA (anti-coagulant) that simultaneously prevents coagulation and stabilizes human WB at the time of collection.

TVT samples were thawed at +37 °C for 30 min for subsequent staining.

### 2.3. Staining Protocols

Fresh and thawed blood samples were lysed and stained with fluorochrome conjugate antibodies as specified in Appendix A, incubated for 30 min at +4 °C, washed once with PBS, filtered with a 70 µm filter, and analyzed on a FACS ARIA III instrument (BD Biosciences).

#### 2.3.1. B Cell Panel

Forward scatter-Area (FSC-A) vs. Side scatter-Area (SSC-A) was used to morphologically identify lymphocytes (Appendix A), whereas singlet lymphocytes were identified by Forward scatter-Height (FSC-H) vs. FSC-A (Appendix A).

B cells were identified as CD19 positive lymphocytes (Appendix A), then classified using different approaches:CD24 vs. CD38 allowed the discrimination of transitional (CD24+ CD38hi), memory (CD24+ CD38−/dim) and naïve-mature (CD24− CD38−/dim) subsets (Appendix A);CD27 vs. IgD identified switched memory (CD27+ IgD−), un-switched memory (CD27+ IgD+), naïve (CD27− IgD+) and CD27− IgD− B cells (Appendix A);IgD vs. CD38, also known as a Bm1-Bm5 classification [14], distinguished six B cell subsets: Bm1 (IgD+ CD38−) mainly virgin-naïve and un-switched memory cells; Bm2 (IgD+ CD38dim) activated-naïve cells; Bm2′ (IgD+ CD38br) pre-germinal center (GC) cells; Bm3-Bm4 (IgD− CD38br) centroblasts and centrocytes present in GC but very low/absent in blood; early Bm5 (IgD− CD38dim) and late Bm5 (IgD− CD38−) memory cells (Appendix A);CD24 vs. CD27 identified CD24+ CD27+ memory cells (Appendix A);CD20 vs. CD38 discriminated plasma blasts/plasma cells (as CD20− CD38hi) (Appendix A);Total IgA+ B cells and total IgD+ B cells were also identified (Appendix A).

#### 2.3.2. CD4 T Cell Panel

FSC-A vs. SSC-A was used to morphologically identify lymphocytes (Appendix A), while singlet lymphocytes were identified by FSC-H vs. FSC-A (Appendix A).

CD4+ T lymphocytes were identified as CD3 (Appendix A) and CD4 positive cells (Appendix A), then characterized based on the expression of the following antigens:CD45RA and CCR7 expression identified CD4+ T cell maturation stages: Naïve (CD45RA+ CCR7+), Central Memory (CM) (CD45RA−- CCR7+), Effector Memory (EM) (CCR7− CD45RA−), and Terminally Differentiated (TD) CD45RA+ CCR7+) (Appendix A) [15];CXCR3 and CCR6 expression discriminated Th1-Th17 (CXCR3+ CCR6+), Th17 (CXCR3− CCR6+) Th1 (CXCR3+ CCR6−), and Th2 (CXCR3− CCR6−) (Appendix A);High expression of CD25 and low expression of CD127 characterized regulatory CD4+ T cells (Tregs) (CD127− CD25 high) (Appendix A). We note that a rigorous Treg identification would require the use of an antibody against the transcription factor FoxP3. However, to avoid fixation and permeabilization steps required to detect FoxP3, we used the surface marker CD127 and CD25, the combination of which provides a reliable inference of Tregs, as identified by FoxP3. Tregs were further subdivided into resting (CD45RA+ CD25+), secreting (CD45RA− CD25+), and activated (CD45RA− CD25++) cells (Appendix A) [16];Among CD4 positive T cells, PD1+ (or CD279+) (Appendix A), CXCR5+ (Appendix A), CCR4+ (Appendix A), ICOS+ (or CD278+) (Appendix A), and CD161+ T cells (Appendix A) were also assessed.

#### 2.3.3. Viability Panel

In flow cytometry, the identification of dead cells is crucial to avoid autofluorescence and unspecific signals. On this purpose, two main viability dye types are available: DNA intercalating dyes and amine-reactive dyes. The intercalating dyes include 7-aminoactinomycin D (7-AAD) and propidium iodide, which non-covalently bind to DNA. Although widely used, they are not suitable for subsequent cell fixation and permeabilization because their fluorescence can be substantially reduced or lost due to their weak binding to the DNA. By contrast, the amine reactive dyes covalently bind proteins, allowing cell fixation and permeabilization without losing cell-viability detection [17]. However, amine-reactive dyes can also bind surface membrane proteins, resulting in a higher fluorescence background compared to DNA intercalators. Thus, if intracellular staining is not needed, DNA dyes are a good choice for viability detection. Among DNA dyes, 7-AAD is more user friendly compared to propidium iodide because it can be stored at +4 °C, is less harmful, does not leach from cells, and has more stable fluorescence [18]. We thus selected 7-AAD for viability detection. Lymphocytes, monocytes, and granulocytes were evaluated by their morphology and by CD45 expression. Cells were then assessed by 7-AAD, which can pass through the compromised cell membrane and bind the DNA. Thus, dead cells will be 7-AAD positive, whereas live cells will be 7-AAD negative. Granulocytes were also evaluated for the granulocyte specific CD66b marker (Appendix A).

#### 2.3.4. M8NK Panel

FSC-A vs. FSC-H was used to discriminate single cells (singlets) (Appendix A). Then, lymphocytes and monocytes were identified by their morphology and CD45 expression (Appendix A). T lymphocytes (CD3+ CD19−) and NK cells (CD3− CD19−) were also discriminated (Appendix A).

T lymphocytes expressing CD8 antigen (cytotoxic T cells) were measured (Appendix A), and then CD27 and CD45RA expression were used to define their maturation status (Appendix A); in detail: naïve (CD45RA+ CD27+), memory cells re-expressing CD45RA (EMRA, CD45RA+ CD27−), Central Memory (CM, CD45RA− CD27−), and Effector Memory (EM, CD45RA− CD27+) CD8+ T cells were assessed [19].

Monocytes (Appendix A) were subdivided into classical (CD14+ CD16−), intermediate (CD14+ CD16+) and non-classical (CD14− CD16+) subsets (Appendix A).

Among NK cells, NK CD56neg (CD16+ CD56−), NK CD56dim (CD16+ CD56dim) and NK CD56bright (CD16− CD56bright) were measured (Appendix A) [20].

### 2.4. Cytometer Settings and Sample Analysis

To maximize fluorescence resolution sensitivity of the analyzer, a cytometer setting setup was generated using the BD Cytometer Setup and Tracking (CS&T) Beads and BD FACSDiva CS&T Module, and specific instrument-setting target values were generated as previously described [1,3].

To ensure measurement reproducibility, CS&T beads were used daily to provide a standardized method to perform quality control of instruments’ optics, electronics, and fluidics, and to adjust fluorescence compensation. The reproducibility of measurements for fresh samples was previously described [1,3].

Compensation was calculated using BD CompBeads (BD Biosciences). These polystyrene microparticles are useful to set accurate single-color compensation for multicolor flow cytometric analyses. After staining with fluorescent antibodies, they generate intense emission signals that, for some antibodies, can be much stronger than the signal intensity obtained by staining cells. In such circumstances, the compensation matrix obtained with the CompBeads may result as excessive if applied to cells. Thus, to ameliorate the compensation matrix generated by the CompBeads, we individually tested each antibody on leukocytes. The improved compensation was then evaluated on samples stained with all antibodies present in the panels and systematically applied to the samples assessed.

Flow cytometric data were analyzed manually by BD FACSDiva™ software (BD Biosciences).

### 2.5. Data Analysis

B, CD4 T cell, and M8NK panels were applied to five samples and performed seven days after the day of freezing, whereas the Viability panel was assessed on three samples 18 months after the day they were frozen (Table 1). The parallel assessment of the five methods would be technically challenging, and thus, we carried out a maximum of two freezing methods at the same time. Because the fresh samples used as reference changed among the five methods, we normalized each frozen sample to the corresponding fresh sample. Specifically, we compared the immune-cell population frequency derived from a thawed sample (T) with the same cell type assessed in fresh blood (F) of the same individual by using the T/F ratio. T/F higher than one means that the cell frequency was augmented in thawed samples compared to the corresponding fresh sample, whereas T/F smaller than one scored a cell frequency reduced in thawed samples. For example, consider two samples, X and Y, processed freshly and with TVT and HeS methods, respectively. The freshly processed X sample has a B-cell frequency (with respect to lymphocytes) of 8%, which, after thawing with TVT methods, is of 6%, yielding a T/F ratio of 0.75. Y sample has a B-cell frequency of 12%, before and after thawing with HeS method, with a T/F ratio of 1. We can conclude that the B-cell frequency is more reliably preserved by HeS compared to TVT: TVT reduced B-cell frequency by 25%.

Before performing the analyses, we tested all traits for normality using the Kolmogorov–Smirnov test [21]. Because data were normally distributed, we analyzed data using parametric tests. We applied the paired *t*-test to compare the average cell frequencies of frozen samples compared to the corresponding fresh samples, whereas we used one-way analysis of variance (ANOVA), followed by the post hoc Tukey test [22] for pairwise comparison of methods, to compare the T/F ratios among the five methods.

All the analyses were conducted using R software v.4.1.1 (https://cran.r-project.org, accessed on 21 August 2010).

A *p*-value < 0.05 was considered statistically significant, meaning that lower is the *p*-value more significant is the difference between cell-type means in fresh and thawed samples, and consequently, the most reliable method will be the one where differences are small and the *p*-value is the least significant.

## 3. Results

We measured 41 immune traits including 21 B cell and 18 T cell subsets assessed as frequency compared to hierarchically higher cell populations, and monocyte and granulocyte levels were evaluated compared to lymphocytes. Each method was assessed in five thawed samples, seven days after the day they were frozen, and in the corresponding fresh samples.

Only for the FM method, we also measured 11 additional traits, quantified in a dedicated panel.

We also evaluated cell viability in three samples per method, 18 months after the day of freezing (Figure 1).

In the text, we indicate only significant *p*-values derived from the paired *t*-test, used to compare the mean of thawed vs. fresh samples. *p*-values derived from ANOVA, used to compare methods, are shown in Appendix A. Summary statistics are detailed in Appendix A.

### 3.1. B Cell Characterization in Thawed Samples

The 18 B-cell subsets were assessed in thawed and fresh samples by a seven-color antibody panel (Appendix A). We observed that after thawing, B-cell frequencies were quite stable in samples treated with CyD, FM and HeS methods (CyD ratio = 1.08, *p*-value = 2.47 × 10^−3^; FM ratio = 1.08; HeS ratio = 1.15) (Table 2), but showed a greater variation in TVT and SmT samples (TVTs ratio= 1.26; SmT ratio = 1.27) (Figure 2).

IgD+ B-cell frequency, which includes naïve and un-switched memory subsets, was comparable between thawed and fresh samples (SmT ratio = 0.96; CyD ratio = 0.92; FM ratio = 0.97; HeS ratio = 1.04; TVT ratio = 0.94;) (Figure 2, Table 2). In contrast, IgA+ B cells were somewhat reduced in SmT (ratio = 0.85) and FM (ratio = 0.93) -treated blood; more strongly reduced in TVTs- and HeS-treated samples (TVT ratio = 0.40 *p*-value = 1.14 × 10^−3^; HeS ratio = 0.72); and completely lost by the CyD method (Figure 2 and Figure 3).

CD38 expression was well detected in SmT-, HeS-, FM-, and TVT-treated samples, whereas it was strongly reduced using the CyD method, thus affecting the discrimination of Bm1-Bm5 B-cell subsets (Table 2).

Remarkably, the expression of CD27 antigen, a pivotal marker to evaluate B-cell maturation stages, was detectable only using HeS and FM methods (Figure 4, Table 2); it was completely lost using CyD, SmT, and TVT protocols, preventing proper identification of naïve, IgD− CD27−, switched and un-switched memory B-cells, as well as the memory subset CD27+ CD24+. HeS and FM-treated samples showed only slight variations of these subsets with respect to fresh samples.

Overall, the FM method gave B-cell levels most similar to those obtained with fresh blood. Indeed, only 11% of immune traits showed significant differences after thawing; in contrast, SmT, CyD and TVT methods showed significant differences between fresh and frozen blood that exceeded 40%. With the FM method, IgA+ B-cell levels were comparable to fresh-treated samples, whereas cell subsets, such as those expressing CD38 marker, showed a higher frequency in FM-thawed samples (Table 2). Similar results were obtained with the HeS method, although IgA+ B cells were substantially underestimated.

### 3.2. CD4 T Cell Characterization in Thawed Samples

The analysis of 21 CD4 T cell populations in thawed blood samples by a 12-color antibody panel (Appendix A) showed important differences based on the cryopreservation procedure used. Interestingly, we did not observe strong frequency variations of CD4+ lymphocytes among methods, with the ratio T/F ranging from a minimum of 0.90 to a maximum of 1.05 in treated blood (Figure 5, Table 2). We detected a loss of CD45RA and CCR7 in TVT samples, which severely impaired the evaluation of T-cell maturation stages (Figure 6). In contrast, CD45RA and CCR7 expression was maintained in SmT, CyD, FM, and HeS approaches, with only small frequency variations in naïve and memory subsets compared to fresh WB (Table 2).

CXCR3 and CCR6 antigens were strongly affected by cryopreservation: they were undetectable in TVTs-, SmT-, CyD-, and HeS-treated samples, and consequently, the discrimination of Th1, Th2, Th1-Th17, and Th17 populations by CXCR3 and CCR6 was only possible using the FM method (Table 2). However, the latter method caused an increase in Th2 (ratio = 1.43, *p*-value = 3.43 × 10^−3^) and Th17 (ratio = 3.12), and a strong reduction in Th1 frequencies (ratio = 0.23, *p*-value = 1.88 × 10^−3^) (Table 2), probably due to low brightness or partial degradation of the CXCR3 antigen after freezing. Interestingly, the chemokine receptor CXCR5, a critical marker for the identification of circulating follicular helper T cells, was quantifiable only in CyD- and FM-treated samples (Figure 7), and only in the latter method was the CXCR5+ CD4+ T cell frequency comparable to fresh samples (FM ratio = 0.81) (Table 2). PD-1+ CD4+ T cells were detectable in the five methods with a strong reduction in TVT and SmT samples (TVT ratio = 0.60, *p*-value = 5.75 × 10^−3^; SmT ratio = 0.53 *p*-value = 4.64 × 10^−3^) and slight variations in the other three methods (CyD ratio = 1.03; HeS ratio = 0.81; FM ratio = 0.84) (Figure 5, Table 2).

Regulatory T cells (Tregs) were identified by the five cryopreserved methods, although with substantial differences. TVT-treated samples showed Treg frequency comparable to fresh WB (ratio = 1.03); nevertheless, because of the loss of CD45RA, the discrimination of naïve and memory subsets was precluded. Treg frequency was mildly reduced with CyD (ratio = 0.70) and halved with the SmT method (ratio = 0.53, *p*-value = 3.61 × 10^−3^). Interestingly, while providing good identification of Tregs, HeS and FM samples underestimated Treg frequencies by about 30% (HeS ratio = 0.68, FM ratio = 0.69, *p*-value = 7.18 × 10^−3^) (Figure 5, Table 2, Appendix A).

Overall, the loss of markers such as CD45RA, CXCR3, and CCR6 in TVTs, SmT, and CyD samples prevented the evaluation of several T-cell subsets. Thus, we concluded that these methods are not suitable for subsequent immunophenotyping. In fact, they showed significant differences between fresh and frozen samples for more than 70% of measured traits.

Similar to B-cell characterization, FM and HeS treated samples displayed good identification of the majority of CD4 T lymphocyte subsets but with significant differences for about 30% of traits. In particular, the frequency of T cells expressing the CXCR3 antigen was undetectable in HeS and strongly reduced in FM samples (ratio = 0.25, *p*-value = 2.68 × 10^−3^). It seems likely that the expression of this marker is deeply influenced by cryopreservation, and for this reason is difficult to characterize in frozen samples.

### 3.3. Monocytes and Granulocytes Detection in Thawed Samples

To assess whether monocytes (M) and granulocytes (Gr) levels were comparable between frozen and fresh samples, we evaluated them as frequencies with respect to lymphocytes (Ly) as M/Ly and Gr/Ly ratios, by gating cells in an FSC-A vs. CD45 plot. Monocytes and granulocytes were detectable in all methods, with frequencies similar to the corresponding fresh samples. The only exception was observed in HeS and TVT, where the granulocyte frequencies were substantially reduced probably due to granule release and/or granulocyte death that leads to dramatic changes in their morphology (Figure 8, Table 2). Granulocytes were also assessed by CD66b positivity, with comparable results.

### 3.4. Leukocyte Viability

Cryopreserved samples require the careful evaluation of cell viability. We thus assessed viability status 18 months after sample freezing. Three of the five methods (CyD, TVT, SmT) included a fixation step that causes cell death (Figure 9). The viability of the two non-fixative methods (FM and HeS) was assessed by two strategies: morphologically gating lymphocytes (FSC-A vs. SSC-A) and using CD45 marker (FSC-A vs. CD45). The first strategy (FSC-A vs. SSC-A) estimated 98% lymphocytes viability because it removed the majority of dead lymphocytes (due to their reduced dimension with respect to viable cells). The second strategy (FSC-A vs. CD45) showed 82% and 87% lymphocyte viability for HeS and FM, respectively. This is because, unlike the morphological parameter FSC-A, CD45 staining does not allow discrimination between living and dead cells (Figure 9). For monocytes and granulocytes, the same principle applies: a morphological identification allows, with good approximation, the discrimination of viable from dead cells. Similar to lymphocytes, monocytes were more than 98% viable if morphologically gated, whereas gating with CD45 reduced viability to 67% and 85% for FM and HeS methods, respectively. Granulocytes appeared almost completely non-viable with the FM method, whereas with the HeS method, they were 92% viable if morphologically gated and 51% viable if identified by the CD45 marker. This may have been because the sedimentation and subsequent removal of red cells by the HeS method could enhance granulocyte viability.

### 3.5. Further Immune Cell Characterization in FM Treated Bloods

The characterization of B- and CD4 T-cell subsets performed in thawed samples showed the best results with the FM cryopreservation method. For this reason, we selected this protocol to evaluate another 11 immune cell populations, comprising cytotoxic T cells (CD8+) and their maturation stages, NKs, and monocytic cell subsets.

Cytotoxic T cell frequencies were well preserved after thawing (Figure 10A), along with their maturation status, which was assessed by CD45RA and CD27 expression (Table 2).

Among NK cells, CD56dim and CD56br subsets maintained a very similar frequency in thawed samples, with T/F ratios close to 1 (NK CD56dim ratio = 1.00; NK CD56br ratio = 1.05) whereas the CD56neg subset was increased after thawing (ratio = 1.26) (Figure 10B, Table 2).

Finally, among monocytes, non-classical and intermediate subsets were reduced in thawed compared to corresponding fresh samples (ratio= 0.62, *p*-value = 3.91 × 10^−2^; ratio = 0.52, respectively), whereas classical monocytes increased after thawing (Figure 10C).

## 4. Discussion

During the last decade, scientific interest in the human immune system has sharply increased, though progress in flow-cytometry technology has been relatively modest compared to technological improvements in both nucleic acid sequencing applications and metabolomic determinations of large numbers of molecules in a few drops of biological sample. Indeed, 50 years after the first one-parameter–one-sample experimental principle was adopted, flow cytometry has attained a level of only 40 parameters simultaneously assessed to capture the biological variation of immune cells in a single run [23]. This technological limitation further requires special attention to sample preparation and experimental design to ensure robust and replicable data. Regarding sample preparation, the scientific community agrees on the negative effects of sample manipulation, ranking whole-blood analysis as preferable for assessing variation in cell levels among samples [24,25]. This has encouraged investigators to use whole blood instead of PBMCs or purified cell types as the starting material for human immunological studies. In fact, PBMC isolation, despite sample purity and high cell viability after storage at low temperature, is time consuming and accompanied by the loss of neutrophils, thus limiting analyses to monocytes and lymphocytes.

The use of fresh whole blood, however, has the major constraint that it is perishable and difficult to use in extensive immunophenotyping of large cohorts or when longitudinally collected sample sets should be analyzed at the same time to avoid “batch” variability (i.e., from different operators, instruments, reagents, etc.). A methodological strategy to overcome potential handling and analysis biases is the cryopreservation of whole blood. However, research groups have had to face the difficulty of guaranteeing the reliability of the assessed samples.

To address this need, we compared five currently available protocols to stabilize and cryopreserve whole blood, characterizing several leukocyte subsets using flow cytometry in thawed and fresh processed samples. Appendix A briefly describes which methods can be used to measure the cell types evaluated in this work.

We showed that different methods to cryopreserve blood for subsequent immunophenotyping differentially affected the accuracy of immune-cell population measurements in thawed samples, due to the altered expression or degradation of cell markers after thawing. For instance, among B lymphocytes, the expression of the memory-cell marker CD27 was strongly affected by some cryopreservation approaches. Indeed, samples treated with TVTs, SmT, and CyD completely lost CD27 expression, restricting the evaluation of B-cell subsets. In contrast, samples treated with HeS and FM methods preserved the CD27 antigen, enabling the evaluation of maturation stages based on IgD and CD27 expression (naïve, IgD− CD27−, switched and unswitched memory B cells). Among CD4 T cells, CXCR3 and CCR6 were deeply influenced by cryopreservation. In fact, we observed their strongly reduced expression or loss in four of five methods, with the only exception being FM-treated samples, in which we were able to detect these markers, although with profound alteration of their frequencies.

One caveat could be that results might depend in part on the different protocols used to thaw the cryopreserved cells which, although suggested by the manufacturer, could negatively impact cell membrane integrity and subsequent staining efficiency. Importantly, three (CyD, TVT, and SmT) of five freezing methods used are characterized by fixation, which, to varying degrees, can damage the integrity of cells before freezing. This is in accord with observations from previous studies showing that after fixation or permeabilization treatments, the signal intensity of several cell surface markers may be partially lost [26,27,28,29]. A potential limitation of the present work is the small sample size and the limited timeframe during which the samples were cryopreserved before immunophenotyping. Indeed, while cell viability was assessed 18 months after freezing, surface markers were evaluated one week after cryopreservation to avoid changes due to antibody fluorescence decay or major technical changes, such as laser replacement, in the analyzer. A second limitation is the restricted number of cell types dissected, with a major focus on the lymphoid compartment, so that additional studies would be needed to analyze the myeloid lineage.

Overall, taking into consideration quality of the data, speed of the protocol, throughput, and cost of the reagents, the FM method clearly seems the most appropriate approach for blood cryopreservation. The choice can positively impact the effectiveness of studies based on large-scale blood collection performed in hospitals or bioresources characterized by extensive numbers of simultaneously collected samples.

Our results support the use of frozen whole blood for large-scale immunophenotyping as an acceptable substitute for analysis of fresh samples, though fresh samples remain the gold standard whenever possible.

## Figures and Tables

**Figure 1 cells-11-01527-f001:**
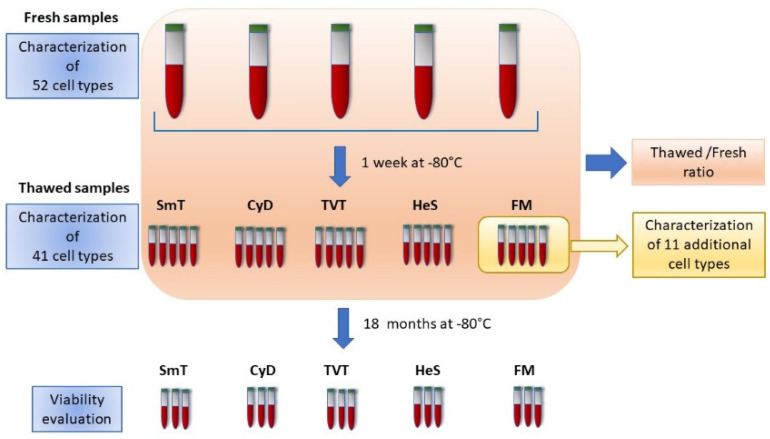
Flow diagram of the study approach. Five blood samples were characterized for 41 immunophenotypes in fresh and matched thawed samples one week after freezing. Eleven additional immunophenotypes were measured in fresh blood and after their freezing by FM method. Thawed and fresh samples were compared by thawed/fresh ratio. Viability was assessed in three samples per method 18 months after storage. SmT: Smart Tubes; CyD: CytoDelics; FM: Freezing Mix; HeS: HetaSep; TVT: TransFix/EDTA Vacuum Tubes.

**Figure 2 cells-11-01527-f002:**
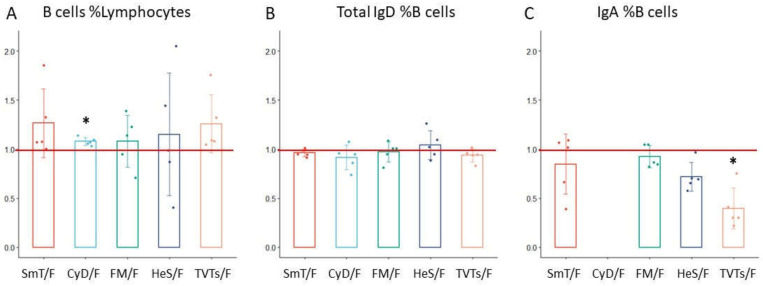
Comparison of B-cell subsets using the five cryopreservation methods assessed. The cell frequency variation in thawed (T) with respect to fresh (F) sample (represented as T/F ratio) is shown for each cryopreservation method. The red line indicates the ideal ratio of 1, meaning no cell frequency variation between cryopreserved and fresh whole blood. The variation of (**A**) B cells, (**B**) IgD positive B-cells and (**C**) IgA positive B-cells were represented as mean values of 5 samples ± standard deviation. * *p*-values from the paired *t*-test ˂ 0.05.

**Figure 3 cells-11-01527-f003:**
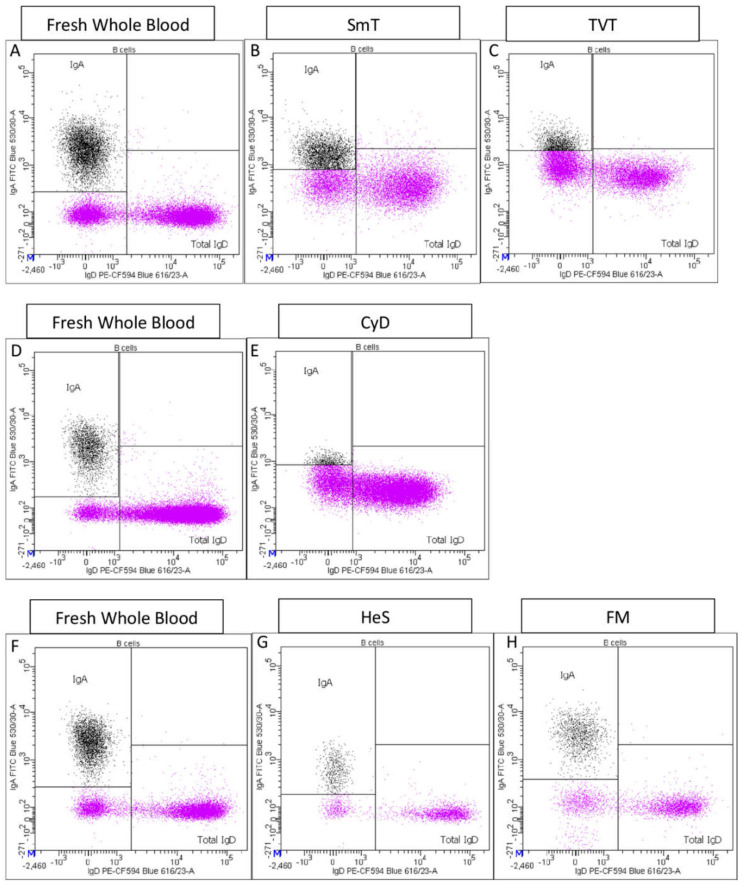
Representative IgA staining of fresh whole blood and thawed samples. The corresponding fresh whole blood is shown on the same line for each freezing method: fresh sample (**A**) is the control for SmT (**B**), and TVT (**C**) methods; fresh sample (**D**) is used as the control for CyD (**E**) method, and fresh sample (**F**) for HeS (**G**) and FM (**H**) methods.

**Figure 4 cells-11-01527-f004:**
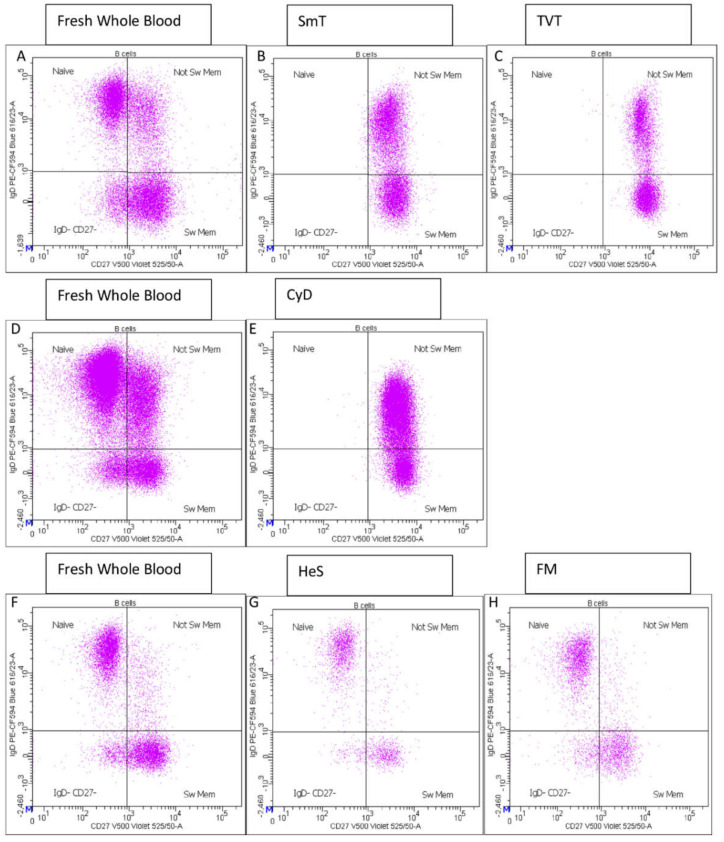
Representative CD27 staining of fresh whole blood and thawed samples. The corresponding fresh whole blood is shown on the same line for each freezing method: fresh blood (**A**) is the control for SmT (**B**), and TVT (**C**) methods; fresh sample (**D**) is used as the control for CyD (**E**) method, and fresh sample (**F**) for HeS (**G**) and FM (**H**) methods.

**Figure 5 cells-11-01527-f005:**
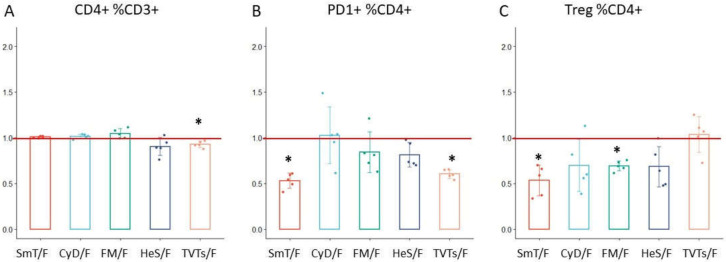
Comparison of CD4 T-cell subsets using the five cryopreservation methods. The cell frequency variation in thawed (T) with respect to fresh (F) sample (represented as T/F ratio) is shown for each cryopreservation method. The red line indicates the ideal ratio of 1 (i.e., with no cell frequency variation between cryopreserved and fresh whole blood). The variation of (**A**) CD4 T cells, (**B**) PD1+ CD4 T cells, and (**C**) Tregs were represented as mean values of 5 samples ± standard deviation. * *p*-values from the paired *t*-test ˂ 0.05.

**Figure 6 cells-11-01527-f006:**
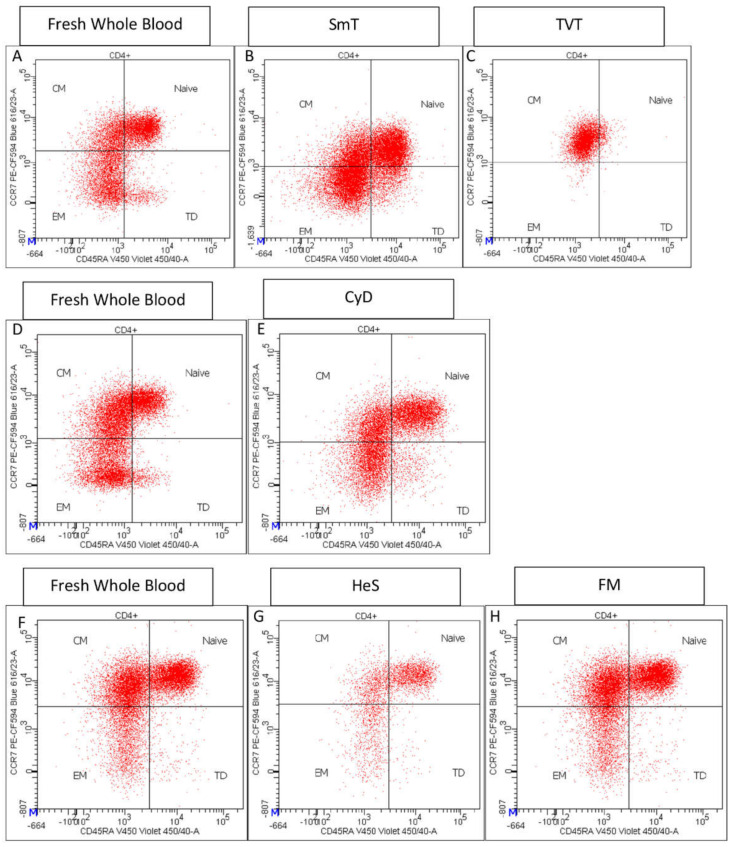
Representative CD45RA vs. CCR7 staining of fresh whole blood and thawed samples. The corresponding fresh whole blood is shown in the same line for each freezing method: fresh blood (**A**) is the control for SmT (**B**), and TVT (**C**) methods; fresh sample (**D**) is used as the control for CyD (**E**) method, and fresh sample (**F**) for HeS (**G**) and FM (**H**) methods.

**Figure 7 cells-11-01527-f007:**
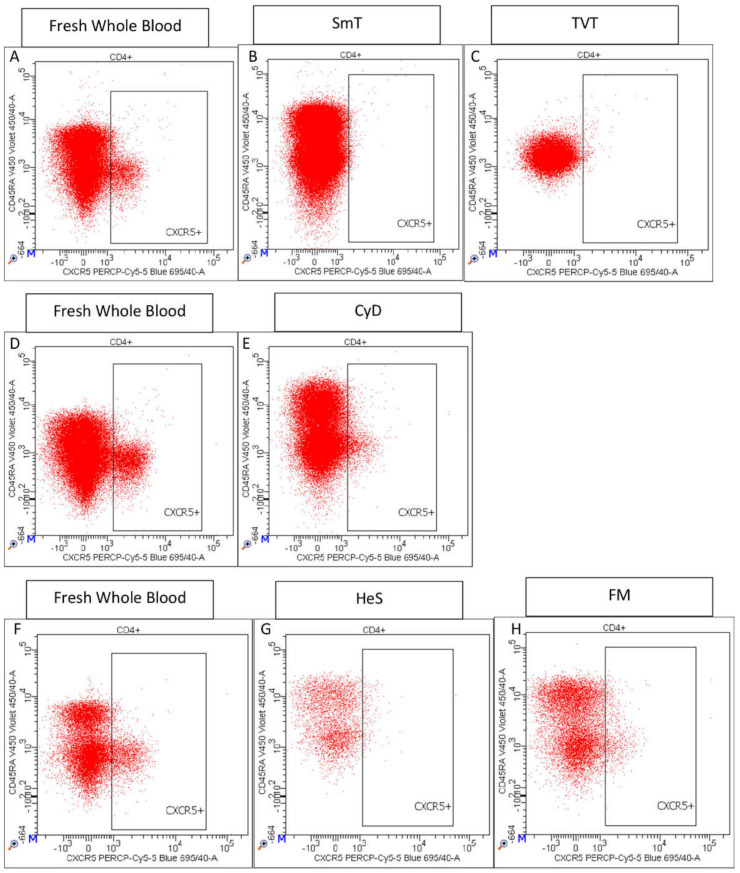
Representative CXCR5 staining of fresh whole blood and thawed samples. The corresponding fresh whole blood is shown on the same line for each freezing method: fresh blood (**A**) is the control for SmT (**B**), and TVT (**C**) methods; fresh sample (**D**) is used as the control for CyD (**E**) method, and fresh sample (**F**) for HeS (**G**) and FM (**H**) methods.

**Figure 8 cells-11-01527-f008:**
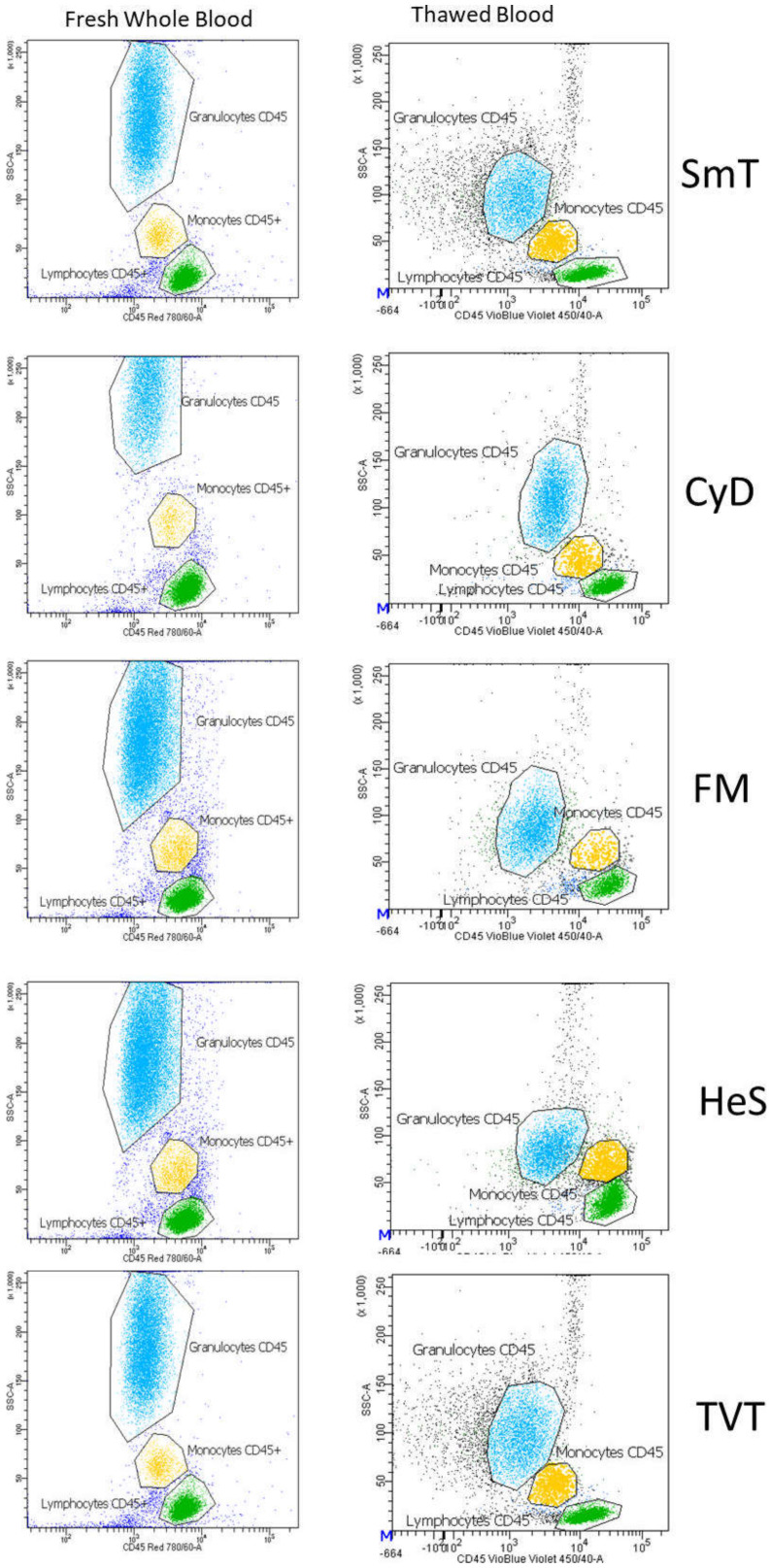
Representative leukocyte staining of fresh and thawed samples. Granulocytes CD45 (light blue), monocytes CD45 (yellow), and lymphocytes CD45 (green) are represented. Each freezing method is indicated on one line; the corresponding fresh whole blood is shown at the left.

**Figure 9 cells-11-01527-f009:**
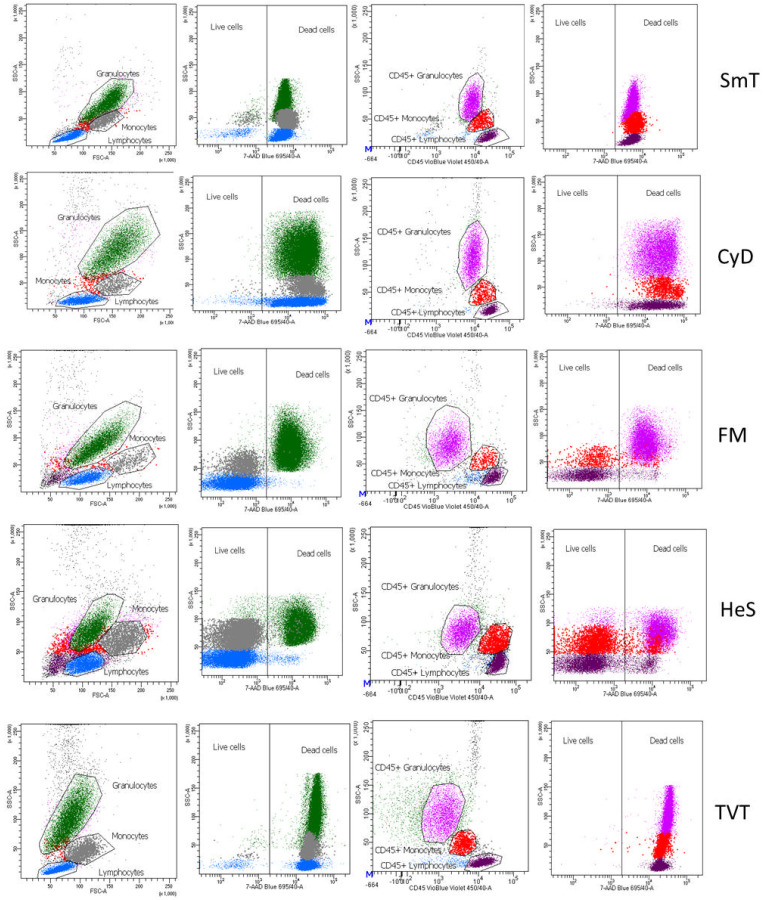
Cell viability after cryopreservation. Leukocyte viability was determined by 7-aminoactimicyn-D negativity in morphologically gated granulocytes (green), monocytes (gray), and lymphocytes (blue); and in CD45+ granulocytes (pink), CD45+ monocytes (red), and CD45+ lymphocytes (purple). A representative sample staining for each method is shown.

**Figure 10 cells-11-01527-f010:**
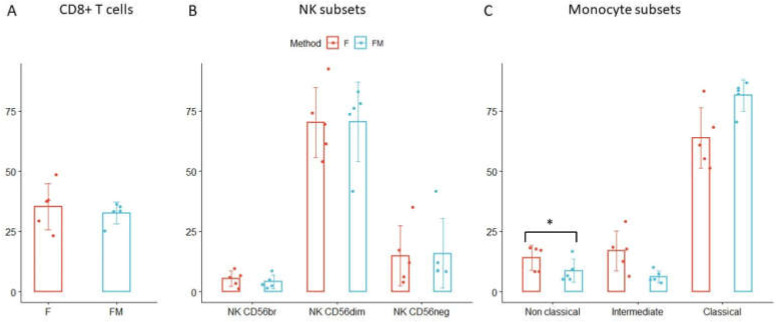
Comparison of cytotoxic T cells, NKs, and monocytes in FM treated and fresh samples. The comparison of cell frequency variation in fresh (F) and thawed samples treated with freezing mix (FM) method is shown. (**A**) CD8+ T cells, (**B**) NK subsets and (**C**) monocyte subsets were represented as mean values of five samples ± standard deviation. * *p*-values from the paired *t*-test ˂ 0.05.

**Table 1 cells-11-01527-t001:** Experimental workflow. Individuals (1–22) are grouped according to the cryopreservation method used and the multicolor panel performed: B, CD4, Viability (Vi) or M8NK (Monocytes-CD8-Natural Killer) panels. Fresh sample staining was performed. The acronyms for the five methods are: SmT (Smart Tubes); CyD (CytoDelics); FM (Freezing Mix); HeS (HetaSep); TVT (TransFix/EDTA Vacuum Tubes).

Methods						
Individuals	Fresh	SmT	CyD	FM	HeS	TVT
1	B-CD4	B-CD4			B-CD4-Vi	B-CD4
2	B-CD4	B-CD4-Vi			B-CD4-Vi	B-CD4-Vi
3	B-CD4	B-CD4			B-CD4-Vi	B-CD4
4	B-CD4	B-CD4-Vi			B-CD4	B-CD4-Vi
5	B-CD4	B-CD4-Vi			B-CD4	
6	B-CD4		B-CD4-Vi			
7	B-CD4		B-CD4-Vi			
8	B-CD4		B-CD4-Vi			
9	B-CD4		B-CD4			
10	B-CD4		B-CD4			
11	B-CD4			B-CD4-Vi		
12	B-CD4			B-CD4-Vi		
13	B-CD4			B-CD4-Vi		
14	B-CD4					
15	B-CD4					
16	M8NK			B-M8NK		
17	M8NK			B-M8NK		
18	CD4			CD4		
19	CD4			CD4		
20	M8NK			M8NK		
21	M8NK			M8NK		
22	M8NK			M8NK		

**Table 2 cells-11-01527-t002:** Thawed/fresh frequency ratios of the immune cells assessed. From left to right are indicated: the cell-type measured; the panel name (Vi = viability panel); the mean; and the standard deviation (±StD) of T/F ratios (r) for the five methods used. (N/A= not assessable).

Trait Name	Panel Name	rSmT		rCyD		rFM		rHeS		rTVTs	
		Mean	±Std	Mean	±Std	Mean	±Std	Mean	±Std	Mean	±Std
B cells %Lymphocyte	B cell	1.27	0.35	1.08	0.04	1.08	0.26	1.15	0.62	1.26	0.30
Total IgD %B cell	B cell	0.96	0.04	0.92	0.12	0.97	0.10	1.04	0.15	0.94	0.07
IgD− CD38br %B cell	B cell	0.56	0.15	N/A	N/A	1.24	0.78	2.25	1.06	0.54	0.33
IgD− CD38dim %B cell	B cell	1.57	0.33	N/A	N/A	0.97	0.49	1.12	0.45	2.65	0.34
IgD− CD38− %B cell	B cell	0.73	0.13	N/A	N/A	1.11	0.27	0.89	0.29	0.36	0.12
IgD+ CD38br %B cell	B cell	0.62	0.39	N/A	N/A	0.73	0.25	1.39	0.80	0.24	0.11
IgD+ CD38dim %B cell	B cell	1.31	0.07	N/A	N/A	0.90	0.30	1.16	0.38	1.47	0.11
IgD+ CD38− %B cell	B cell	0.75	0.20	N/A	N/A	1.30	0.42	0.82	0.35	0.27	0.12
Naive %B cell	B cell	N/A	N/A	N/A	N/A	0.92	0.15	1.03	0.19	N/A	N/A
Not Sw Mem %B cell	B cell	N/A	N/A	N/A	N/A	1.06	0.21	1.01	0.26	N/A	N/A
IgD− CD27− %B cell	B cell	N/A	N/A	N/A	N/A	1.11	0.29	0.81	0.25	N/A	N/A
Sw Mem %B cells	B cell	N/A	N/A	N/A	N/A	1.04	0.18	1.07	0.33	N/A	N/A
Plasma cells %B cell	B cell	0.90	0.96	N/A	N/A	2.52	1.00	2.26	1.22	1.58	1.04
Memory %B cell	B cell	0.67	0.19	0.82	0.10	0.79	0.14	0.89	0.20	0.49	0.07
Naive-Mature %B cell	B cell	1.22	0.06	1.37	0.53	1.13	0.09	0.99	0.12	1.37	0.15
CD24+ CD27+ %B cell	B cell	N/A	N/A	N/A	N/A	0.88	0.24	0.91	0.23	N/A	N/A
Transitional %B cell	B cell	0.85	0.26	0.78	0.03	0.95	0.18	1.66	0.11	0.90	0.24
IgA %B cell	B cell	0.85	0.31	N/A	N/A	0.93	0.11	0.72	0.15	0.40	0.21
CD3+ %Lymphocyte	CD4	1.04	0.05	1.08	0.05	1.19	0.17	0.89	0.12	1.00	0.04
CD4+ %CD3+	CD4	1.01	0.01	1.02	0.02	1.05	0.05	0.90	0.10	0.93	0.04
Treg %CD4+	CD4	0.53	0.17	0.70	0.29	0.69	0.05	0.68	0.22	1.03	0.19
resting %Treg	CD4	1.10	0.28	3.04	2.03	2.30	1.26	1.90	0.79	N/A	N/A
activated %Treg	CD4	0.94	0.27	0.99	0.36	0.80	0.16	0.92	0.12	N/A	N/A
secreting %Treg	CD4	1.08	0.25	0.86	0.13	1.11	0.10	1.00	0.11	N/A	N/A
Th17 %CD4+	CD4	N/A	N/A	N/A	N/A	3.12	4.76	N/A	N/A	N/A	N/A
Th1-Th17 %CD4+	CD4	N/A	N/A	N/A	N/A	0.93	1.13	N/A	N/A	N/A	N/A
Th2 %CD4+	CD4	N/A	N/A	N/A	N/A	1.43	0.20	N/A	N/A	N/A	N/A
Th1 %CD4+	CD4	N/A	N/A	N/A	N/A	0.23	0.28	N/A	N/A	N/A	N/A
CXCR3 %CD4+	CD4	N/A	N/A	N/A	N/A	0.25	0.30	N/A	N/A	N/A	N/A
CCR6 %CD4+	CD4	N/A	N/A	N/A	N/A	1.92	3.41	N/A	N/A	N/A	N/A
CCR4 %CD4+	CD4	0.15	0.03	0.77	0.60	0.92	0.11	1.12	0.32	0.23	0.04
CM %CD4+	CD4	1.17	0.12	0.90	0.08	1.06	0.14	0.90	0.21	N/A	N/A
Naive %CD4+	CD4	1.06	0.03	1.35	0.12	1.29	0.22	1.39	0.27	N/A	N/A
EM %CD4+	CD4	0.76	0.07	0.80	0.15	0.69	0.14	0.84	0.32	N/A	N/A
TD %CD4+	CD4	1.12	0.26	1.09	0.33	1.01	0.55	1.00	0.28	N/A	N/A
CXCR5 %CD4+	CD4	N/A	N/A	0.36	0.28	0.81	0.57	N/A	N/A	N/A	N/A
ICOS+ %CD4+	CD4	0.56	0.21	0.47	0.15	0.79	0.36	0.97	0.39	0.68	0.18
PD1+ %CD4+	CD4	0.53	0.08	1.03	0.31	0.84	0.22	0.81	0.13	0.60	0.05
CD161+ %CD4+	CD4	0.44	0.12	N/A	N/A	0.74	0.23	0.68	0.08	0.95	0.07
CD8+ %CD3+	M8NK	-	-	-	-	0.96	0.20	-	-	-	-
Non classical %Monocytes	M8NK	-	-	-	-	0.62	0.20	-	-	-	-
Intermediate %Monocytes	M8NK	-	-	-	-	0.52	0.43	-	-	-	-
Classical %Monocytes	M8NK	-	-	-	-	1.32	0.29	-	-	-	-
EMRA CD8+ %CD8+	M8NK	-	-	-	-	1.18	0.45	-	-	-	-
Naive CD8+ %CD8+	M8NK	-	-	-	-	1.06	0.25	-	-	-	-
CM CD8+ %CD8+	M8NK	-	-	-	-	1.12	0.50	-	-	-	-
EM CD8+ %CD8+	M8NK	-	-	-	-	1.07	0.51	-	-	-	-
NK CD56br %NKs	M8NK	-	-	-	-	1.05	0.90	-	-	-	-
NK CD56dim %NKs	M8NK	-	-	-	-	1.00	0.18	-	-	-	-
NK CD56neg %NKs	M8NK	-	-	-	-	1.26	0.65	-	-	-	-
M/Ly	Vi	1.19	0.05	1.20	0.19	0.96	0.35	1.06	0.13	0.95	0.04
Gr/Ly	Vi	1.23	0.18	1.17	0.05	1.06	0.22	0.35	0.12	0.78	0.33

## Data Availability

Not applicable.

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
