# Peer review of "Comparison of Whole Blood Cryopreservation Methods for Extensive Flow Cytometry Immunophenotyping"

_cells, 2022, doi:10.3390/cells11091527_

Round 1

Reviewer 1 Report

The aim of the study was to compare the whole blood cryopreservation methods for extensive flow cytometry immunophenotyping.  The approach of the study appears very original. The contents of the manuscript are quite interesting by his methodology and through the tools of quantification used. The very informative figures are used. The manuscript reads smoothly and is easy to understand. The aims, scope, and results of the study are clearly stated. I have very much enjoyed reading this paper.

The cryopreservation is very important method especially when study require the collection and analysis of many thousands of individuals. The whole blood sample cryopreservation allows to storage all types of leukocytes in the very fast way.

Comments:

I will recommend to expand the results obtained by other authors ( ref. 5&6).

I will recommend to add the table which briefly compare which method is suitable for which evaluation.

Reviewer 2 Report

By assessing the amounts of 41 immune cell types, researchers compared the quality of immunophenotypic data acquired from fresh blood to those obtained from five cryopreservation procedures. The study is beneficial and deserves publication.

In abstract, describe briefly the cryopreservation methods.

The statistical analysis needs revision, why you selected one-way ANOVA. Did you tested the normal distribution of data?

Discuss the potential limitations that may exist.

Justify the selected sampling and testing durations. Discuss the potential chanes after longer sampling preservation.

The manuscript did not sufficiently discussed the results in the light of recent findings e.g. PMID: 33544978

The manuscript needs extensive language revision, few examples are below

Line 81 manufacturer instructions

Line 115 stored at

Lines 203-205 rephrase

Line 228 reduces B cell frequencies by

Line 229 To identify what method performs

Line 235 P-value

Reviewer 3 Report

This is an interesting manuscript describing how to mitigate the challenges in characterizing the immune cells using flow cytometry. The methods and results have been described well. I have following minor concerns

  1. Please define/expand the medium TVT, SmT, CyD, HeS or FM in Table 1 and the catalog # in subsections 2.
  2. 7-aminoactinomycin D was used for live-Dead sorting. Please mention as it will help researchers early in their career to choose ideal marker. There are various kits available for this (for e.g., https://www.thermofisher.com/us/en/home/references/protocols/cell-and-tissue-analysis/protocols/live-dead-fixable-dead-cell-stains.html) , why 7-AAD was used, please mention the rationale.
  3. Along with the description for sorting strategy for various immune cells, including a flow-chart of the strategy will be more informative.
  4. In ST1, it will be good to include the catalog number for each antibody
  5. Please define all abbreviations at their first appearance

Round 2

Reviewer 2 Report

Accept